

# Validation of PozQoL scale in Turkish population living with HIV: a cross-cultural adaptation study

Sabri Atalay[1], Zeynep Sedef Varol[2], Sarp Singil[1] and Ufuk Sönmez[3]

[1] Department of Infectious Diseases and Clinical Microbiology, University of Health Sciences Tepecik Training and Research Hospital, Izmir, Turkey
[2] Department of Public Health, Division of Epidemiology, Dokuz Eylül University Faculty of Medicine, Izmir, Turkey
[3] Department of Infectious Diseases and Clinical Microbiology, University of Health Sciences Bozyaka Training and Research Hospital, Izmir, Turkey

## ABSTRACT

**Background:** The increasing number of people living with HIV requires a simple and easy-to-use quality of life (QoL) scale for people living with HIV (PLWH). This study aims to adapt the PozQoL scale into Turkish and assess its reliability and validity for the PLWH population in Turkey.

**Methods:** Translation-back-translation methodology was employed, and face-to-face interviews were conducted with 130 patients using the PozQoL, socio-demographic, and clinical data questionnaire. Exploratory Factor Analysis (EFA) and Confirmatory Factor Analysis (CFA) were utilized to identify the underlying factor structure and examine the validity of the measurement model, respectively. Cronbach's alpha and intraclass correlation coefficients (ICCs) were used to assess internal consistency and test-retest reliability, respectively.

**Results:** EFA revealed four factors with an eigenvalue of 0.88, explaining 62.1% of the cumulative variance. CFA indicated that the four-factor solution achieved good levels of fit. The total Cronbach's alpha was 0.81, indicating high internal consistency. The ICC for the total score was 0.92 (95% confidence interval (CI) [0.90–0.94]; $p < 0.05$), demonstrating high test-retest reliability.

**Conclusion:** The Turkish version of the PozQoL was found to be a valid and reliable tool for assessing the health-related QoL of PLWH in Turkey.

## INTRODUCTION

According to the Joint United Nations Programme on HIV/AIDS (UNAIDS), a total of 84.2 million people have been infected since the beginning of the HIV/AIDS pandemic, of whom 40.1 million have died. Today, 38.4 million people are living with HIV, yet AIDS-related deaths are the lowest ever recorded. Most people living with HIV (85%) know their HIV status and 28.7 million people are receiving HIV treatment (*UNAIDS, 2022*). As a result, HIV infection has become a chronic condition in which the life expectancy is almost the same as that of HIV-negative people (*Kall et al., 2021*). However, people living with HIV still face many physical, mental and social problems. These include

Corresponding author
Sabri Atalay, drsatalay@yahoo.com

depression, anxiety, financial problems, chronic illnesses, stigma and discrimination. Therefore, adding quality of life (QoL) as an unmet need in HIV management has come to the fore (*Lazarus et al., 2016*). The QoL of people living with HIV (PLWH) is an important measure of the impact of the disease and its treatment on their physical, emotional, and social well-being. There is growing interest in using patient-reported outcome measures (PROMs) such as EuroQoL, PROQOL-HIV and WHOQOL-HIV-BREF to measure the health-related quality of life (HRQoL) of PLWH (*Kall et al., 2020*).

There has been a significant increase in the number of HIV/AIDS cases in Turkey over the past few years. The total number of cases reached 39,437 people from 1985 to 08 November 2023 (*Republic of Türkiye Ministry of Health, 2023*). Therefore, there is also a need for a reliable quality of life scale that can be used in our country as the number of people living with HIV increases. One of these scales is the POZQoL scale developed by the Australian Research Centre in Sex, Health and Society (ARCSHS) and project partners (*Brown et al., 2018*). The inter-item reliability (assessed by Cronbach's alpha) was $\alpha = 0.95$ (95% CI [0.93–0.96]) in a validity-reliability study with the original English text of the PozQoL scale, indicating excellent internal consistency. The reliabilities for specific subdimensions were: health-related-$\alpha = 0.91$ (95% CI [0.89–0.93]); psychological-$\alpha = 0.91$ (95% CI [0.89 to 0.93]); social-$\alpha = 0.82$ (95% CI [0.77 to 0.86]); and functional-$\alpha = 0.87$ (95% CI [0.84–0.90]). Although the sensitivity and reliability studies of the POZQoL scale have been conducted on the English version, there are versions of the scale adapted to 15 world languages (*PozQoL, 2024*). As this scale is short, simple, easy to use, it was considered suitable for evaluating the QoL of PLWH in our country. However, its applicability in other countries and cultures needs to be evaluated. The main objective of this study is to validate and adapt the PozQoL scale for Turkish PLWH, and to determine the reliability and validity of the scale in assessing the QoL of this population in Turkey.

## MATERIALS AND METHODS

### PozQoL scale

The PozQoL scale was developed through a structured literature review and consultation with PLWH peer organizations to identify the key domains of health related quality of life (HRQoL). An initial pool of over 100 candidate items was developed and pre-tested for face and content validity through an online survey with a panel of experts. The pool of items was then reduced to 64. An online survey of adult Australians living with HIV was conducted to develop and test PozQoL psychometrically, including a 1-month follow-up survey to examine test-retest reliability. The study found that the PozQoL scale is a reliable and valid measure of HRQoL for PLWH, with a Cronbach's alpha score of 0.91 indicating high internal consistency. The questionnaire had four factors: psychological, physical, social, and functional wellbeing. The study used Exploratory Factor Analysis (EFA) and Confirmatory Factor Analysis (CFA) to assess the questionnaire's goodness of fit and identified its underlying factor structure. Overall, the PozQoL questionnaire had good psychometric properties and could be used to assess the QoL of PLWH in Australia (*Brown et al., 2018*).

## Study design and sampling method

According to previous research on questionnaire adaptation, it is recommended that the sample size be 5–10 times the number of items in the questionnaire (*Tabachnick & Fidell, 2020*). Given that the PozQoL scale comprises 13 items, we recruited a total of 130 patients living with HIV for this study. Participants were selected from those who presented at the Infectious Diseases and Clinical Microbiology Clinic of Izmir Tepecik Training and Research Hospital between October 26th, 2022, and February 15th, 2023.

## Translation and adaptation of the PozQoL

The translation-back-translation methodology was used to adapt the PozQoL scale following the Guidelines for Translating and Adapting Tests by the *International Test Commission (2017)*. Two bilingual translators proficient in both English and Turkish independently translated the English version of PozQoL into Turkish, after which two other translators performed back-translations independently. The resulting Turkish version was reviewed and modified by a team of one Epidemiologist, one Public Health Specialist and three Infectious Diseases and Clinical Microbiology specialists to ensure its suitability for the specifics of Turkish medical and caring systems and culture. After an expert panel review, a draft of the Turkish version of PozQoL was formulated. The revised Turkish version was tested with a small group of 20 individuals who were not included in the final study. These patients did not provide any suggestions or corrections about the questionnaire during this stage. Finally, the revised Turkish version was developed into the final text of the Turkish version of PozQoL scale.

## Data collection

Data collected in face-to-face interviews using PozQoL, socio-demographic and clinical data questionnaire.

## Analyse

The study utilized an EFA to identify the underlying factor structure, and a CFA was conducted to test the final model using several goodness-of-fit measures. The final scale and subdimensions' inter-item reliability were assessed using Cronbach's alpha, and the temporal stability (test-retest reliability) of the scale was evaluated using intraclass correlation coefficients (ICCs) among the participants who completed the follow-up sample.

Descriptive characteristics of the participants were summarized using number, percentage, mean, and standard deviation values, and the data analysis was conducted using IBM SPSS Statistics for Windows, version 29.0, and CFA was performed with SPSS AMOS for Windows, version 29.0 software.

## Ethical considerations

The study received written authorization to adapt the PozQoL into Turkish through email communication with project team at La Trobe University where they developed PozQoL. The authors have permission to use this instrument from the copyright holders
(*Brown et al., 2018*). Additionally, ethical clearance to conduct the research was obtained from the ethics committee of the Tepecik Training Hospital (approval date/number: 15.09.2022/2022/09-31) and the administration of the hospital where the participants were recruited. Informed written consent from participants was taken.

## RESULTS

The study group consisted of 119 (91.5%) male subjects, of whom 43 (33.1%) were married and 94 (72.3%) had a secondary school education or less. Participants ranged in age from 20 to 76 years, with a mean age of 40.2 ± 12.2 years. The mean duration of HIV diagnosis was 49.7 ± 45.9 months (min–max: 2 to 216 months), while the mean duration of antiretroviral treatment was 45.5 ± 40.0 months (min–max: 2 to 216 months).

### Construct validity

#### *Explanatory factor analysis*

The sampling adequacy was found to be satisfactory with a Kaiser-Mayer-Olkin (KMO) value of 0.81 and a significant Bartlett's Test value of 464.1 ($p < 0.001$). Factor analysis was performed using principal component analysis and direct oblimin rotation. Explanatory Factor Analysis (EFA) revealed four factors with an eigenvalue of 0.88, which explained 62.1% of the cumulative variance. The rationale for excluding an eigenvalue of one was informed by the visual inspection of the scree plot graph, which revealed a four-factor structure. The loading weights, obtained with the EFA, are shown in Table 1. As seen in Table 1, five items loaded on Factor 1, four items loaded on Factor 2, two items loaded on Factor 3, two items loaded on Factor 4.

Factor 1 consists of five items: 9, 10, 11, 12, and 6. Specifically, items 9 and 11 belong to the social subdimension, items 10 and 6 belong to the functional subdimension, and item 12 belongs to the health subdimension. The factor loadings of the items under Factor 1 are high, whereas the factor loading of item 6 is at a moderate level. Furthermore, item 6 has a factor loading distribution with values that differ by less than 0.1 under Factor 3 and Factor 4. On the other hand, items 13, 1, 5, and 8 belong to the psychological, and their factor loading under Factor 2 is greater than 0.6. Items 2 and 7 in the health subdimension are under Factor 3; however, the factor loading of item 2 is at the borderline level. Item 3 in the social subdimension and item 4 in the functional subdimension are under Factor 4, and their factor loadings are good. The EFA reveals that the items under the health subdimension are under Factor 3, except for item 12. Four items in the psychological subdimension are under Factor 2. The items in the social and functional subdimensions are under Factor 1 and Factor 4, respectively.

At this stage, only the items in the psychological subdimension are under Factor 2. Other factors have not been named, and the original subdimension model of the scale has been tested using CFA.

### Confirmatory factor analysis

CFA utilized to examine the validity of the measurement model for a set of items. The analysis was performed using the maximum likelihood procedure on the covariance

**Table 1 Factor structure of the Turkish version of the PozQoL.**

| Items | Subdimension items of the PozQoL scale | | | |
|---|---|---|---|---|
| | Factor 1 | Factor 2 | Factor 3 | Factor 4 |
| 9) I am afraid that people may reject me when they learn I have HIV | *0.75* | 0.02 | 0.09 | 0.01 |
| 10) Managing HIV wears me out | *0.72* | 0.20 | 0.32 | 0.06 |
| 11) I feel that HIV limits my personal relationships | *0.72* | 0.05 | 0.18 | 0.27 |
| 12) I fear the health effects of HIV as I get older | *0.62* | 0.01 | **0.45** | 0.04 |
| 6) Having HIV limits my opportunities in life | *0.42* | 0.07 | **0.33** | **0.37** |
| 13) I am optimistic about my future | 0.07 | *0.81* | 0.03 | 0.11 |
| 1) I am enjoying life | 0.22 | *0.73* | 0.22 | 0.14 |
| 5) I feel good about myself as a person | 0.04 | *0.70* | 0.27 | 0.01 |
| 8) I feel in control of my life | 0.02 | *0.67* | **0.46** | 0.01 |
| 7) I worry about the impact of HIV on my health | **0.35** | 0.12 | *0.71* | 0.21 |
| 2) I worry about my health | 0.17 | 0.16 | *0.58* | **0.36** |
| 3) I lack a sense of belonging with people around me | 0.11 | 0.07 | 0.20 | *0.79* |
| 4) I feel that HIV prevents me from doing as much as I would like | 0.43 | 0.09 | 0.01 | *0.70* |

Note:
 Item distributions with factor loadings above 0.3 are shown in bold; the factor structure of each item is shown in bold and italicised.

matrix, and the items were constrained to each load on one factor. Goodness-of-fit was assessed using a variety of fit indices, including the model chi-square/df, comparative fit index (CFI), standardized root-mean-square residual (SRMR), and root-mean-square error of approximation (RMSEA).

The study employed the guidelines for optimal fit proposed by *Lt & Bentler (1999)*, which recommended chi-square/df ratio between 1 and 3 is considered acceptable, CFI > 0.95, SRMR < 0.08, and RMSEA < 0.06. The findings indicated that the four-factor solution achieved good levels of fit, with a model chi-square of $X^2$/df = 1.48, a CFI of 0.93, an SRMR of 0.062, and an RMSEA of 0.06 (*Lt & Bentler, 1999*; *McDonald & Ho, 2002*).

## Reliability

### Internal consistency

Internal consistency was assessed through Cronbach's alpha scores. According to the statement, the reliability level anticipated for the research-useable measurement tools is 0.70 or higher (*Tabachnick & Fidell, 2020, 2018*). Considering the internal consistency of the subdimension and reliability of the Turkish version of PozQoL, the total Cronbach's alpha was 0.81. Cronbach's alpha values ranged from 0.39 to 0.74 (Table 2). There was no item identified that exhibited a substantial impact on the Cronbach's alpha coefficient upon its removal.

### Interrater reliability

Interrater reliability was evaluated with ICC. The consistency of PozQoL results over time was evaluated by comparing all the participants' initial test results with a retest performed 4 weeks later. The ICC for total score was 0.92 (95% confidence interval (CI) [0.90–0.94]; $p < 0.05$). ICCs ranged from 0.69 to 0.84. Although the Cronbach's alpha of the "social"

**Table 2 Reliability analyses of the Turkish version of the PozQoL.**

|  | Cronbach's alpha | ICC |  |
| --- | --- | --- | --- |
|  |  |  | *p* value* |
| Health | 0.71 | 0.82 | <0.001 |
| Psychological | 0.74 | 0.84 | <0.001 |
| Social | 0.39 | 0.69 | <0.001 |
| Functional | 0.63 | 0.81 | <0.001 |
| TOTAL | **0.81** | **0.91** | **<0.001** |

Notes:
* Correlation is significant at 0.001.
ICC, Intraclass Correlation Coefficient.
The total is shown in bold.

**Table 3 Item scale correlations of the Turkish version of the PozQoL.**

| Items | Health | Psychological | Social | Functional |
| --- | --- | --- | --- | --- |
| Item 1 | 0.17* | **0.68*** | 0.17 | 0.19* |
| Item 2 | **0.76**** | 0.30** | 0.41** | 0.45** |
| Item 3 | 0.28** | 0.17 | **0.55**** | 0.28** |
| Item 4 | 0.40** | 0.19* | 0.52** | **0.74**** |
| Item 5 | 0.22* | **0.74**** | 0.08 | 0.18* |
| Item 6 | 0.46** | 0.20* | 0.38** | **0.81**** |
| Item 7 | **0.85**** | 0.29** | 0.45** | 0.47** |
| Item 8 | 0.34** | **0.76**** | 0.19* | 0.26** |
| Item 9 | 0.37** | 0.09 | **0.71**** | 0.34** |
| Item 10 | 0.49** | 0.28** | 0.45** | **0.73**** |
| Item 11 | 0.54** | 0.17 | **0.75**** | 0.58** |
| Item 12 | **0.78**** | 0.16 | 0.55** | 0.49** |
| Item 13 | 0.16 | **0.79**** | 0.19* | 0.23** |

Notes:
The correlations with strong associations between the items and their corresponding subdimensions were given in bold.
*$p < 0.05$ and **$p < 0.01$.

subdimension was low (0.39), test-retest reproducibility was good according to an ICC of 0.69 (Table 2) (*Koo & Li, 2016*).

### *Item-scale correlations*

Item-scale correlations were computed to assess the strength of the relationship between each item and its corresponding scale. The results of the item-scale correlations are presented in Table 3. As shown, all items had positive and significant correlations with their respective scales, indicating good internal consistency. The correlations ranged from 0.55 to 0.85, indicating strong associations between the items and their corresponding subdimensions. These findings suggest that the items within each scale are measuring the same underlying construct and are consistent with the overall reliability of the scales.

### Subdimension scores of the PozQoL

The mean and standard deviation scores for the PozQoL subdimensions were as follows: total score 47.1 ± 10.9, health subdimension 10.2 ± 3.7, psychological subdimension 15.5 ± 4.2, social subdimension 10.5 ± 3.1, and functional subscale 10.8 ± 3.5.

## DISCUSSION

The PozQoL was found to be a valid and reliable tool to assess the health quality of PLWH in Turkey.

Based on the EFA, the final Turkish version of PozQoL was found to be consistent with the original PozQoL. Specifically, the Turkish PozQoL was found to have four subdimensions, mirroring those of the original version. Recent statistical studies have recommended against removing items with factor loadings exceeding 0.30 from questionnaires (*Tabachnick & Fidell, 2020*). In this study, all items demonstrated factor loadings greater than 0.30, indicating that no items were eliminated from the questionnaire.

In the CFA conducted during the validity-reliability study of the scale, which is currently in use and has been translated into 16 languages, the model fit indices were good in the Turkish language. However, as CFA was not performed on the different language versions of the scale, no comparison could be made. This adds value to our study. It also differentiates it from other language versions.

Items 3 and 4 were categorized differently as social and functional in the original version of the scale, whereas both of them together in factor 4 were perceived as functional by the patients. Of items 9, 10, 11 and 6, two should have been in the social category (9 and 11) and two in the functional category (6 and 10), all of these items were evaluated under Factor 1. Item 12 was a health-related, while it was evaluated with the social and functional items in Factor 1. This may be due to differences in the interpretation of factors such as the perception of health and the definition of symptoms in different countries (*Schmidt & Bullinger, 2003*). Items evaluating psychological factors were found to be compatible with each other.

Differences in language, values, traditions, beliefs and culture have also been found to influence questionnaires in other QoL studies (*Schmidt & Bullinger, 2003*; *Beaton et al., 2000*). Therefore, in addition to accurate translation, adaptation to the culture in which it will be administered is extremely important when developing a questionnaire in another language (*Beaton et al., 2000*). For example, Asian respondents were found to be more likely to agree with the questions asked and to avoid giving extreme answers (*Azocar et al., 2001*). In contrast, there was less item variance in surveys conducted in culturally similar languages (*Schmidt, Mühlan & Power, 2006*). Therefore, it is not easy to decide whether a difference found in the measurements is a real difference or due to different interpretations by people in that country and culture (*Teresi & Fleishman, 2007*). In addition, the questions asked in the questionnaire, the health status and age of the participants and the statistical method applied may also be related to the different results obtained (*Bagheri et al., 2022*).

The construct validity was assessed using CFA and comparison of known groups. CFA is typically utilized for scale development, validity analysis, or to confirm a pre-determined factor structure (*Kline, 2015*). Overall, the study effectively used CFA to validate the measurement model for a set of items. The use of multiple fit indices provided a comprehensive evaluation of model fit, and the findings suggested that the four-factor solution achieved a good level of fit. These results enhance the confidence in the validity of the measurement model and provide a strong foundation for future research utilizing these items. However, it is important to note that the sample size and composition can impact the results of CFA, and additional research may be needed to evaluate the generalizability of these findings to other populations.

The current study assessed the time invariance of the PozQoL through test-retest analysis and its internal consistency through Cronbach's alpha. The results indicated a moderate to excellent agreement between the PozQoL test and retest. The interpretability of the ICC score, which indicates the acceptable or good score, depends on the analysis context and purpose. Typically, an ICC score above 0.75 is regarded as excellent, 0.60–0.74 as good, 0.40–0.59 as fair, and below 0.40 as poor (*Koo & Li, 2016*). In our study, the ICC value was found to be 0.91. This result showed that the exhibits invariance over time and confirmed retest reliability. In the original study, scores on the PozQoL displayed high levels of stability, ICC = 0. 95 (95% CI [0.92–0.97]). Similarly, stable results were obtained on scores for all subdimensions: ICC = 0.91 (95% CI [0.85–0.95])-health concerns; ICC = 0.85 (95% CI [0.74–0.92])–psychological; ICC = 0.83 (95% CI [0.71–0.91])–social; and ICC = 0.89 (95% CI [0.80–0.93])-functional (*Brown et al., 2018*).

Internal consistency is commonly evaluated using Cronbach's alpha coefficient of reliability. A tool is considered moderately reliable when Cronbach's alpha values are between 0.60–0.79, while values in the range of 0.8–1 indicate high reliability (*Tabachnick & Fidell, 2018*). In this study, the scale showed high reliability with a Cronbach's alpha value of 0.81. The Cronbach's alpha values for the subdimensions and were also found to be moderate except social dimension (0.39). Other QoL scales showed similar differences. For example, while five domains in the Chinese version of the WHOQOL-HIV BREF scale showed adequate internal consistency, the spirituality domain had a relatively low Cronbach's alpha coefficient of 0.70. However, the results obtained support the use of this scale (*Zhu, Liu & Qu, 2017*). In the Spanish version of the same scale, the Cronbach's alpha was between 0.61 and 0.81 and was found to be at an acceptable level (*Fuster-RuizdeApodaca et al., 2019*).

Consequently, the Turkish version of PozQoL was found to be valid and reliable. These findings support the use of this tool to measure QoL in Turkish PLWH.

## CONCLUSIONS

The PozQoL was found to be a valid and reliable tool to assess the health quality of PLWH in Turkey. This scale can be used in the assessment of QoL of PLWH in our country and in studies to be conducted in this population.

### Funding

The authors received no funding for this work.

### Competing Interests

The authors declare that they have no competing interests.

### Author Contributions

- Sabri Atalay conceived and designed the experiments, performed the experiments, analyzed the data, prepared figures and/or tables, authored or reviewed drafts of the article, and approved the final draft.
- Zeynep Sedef Varol conceived and designed the experiments, performed the experiments, analyzed the data, prepared figures and/or tables, authored or reviewed drafts of the article, and approved the final draft.
- Sarp Singil conceived and designed the experiments, performed the experiments, analyzed the data, prepared figures and/or tables, authored or reviewed drafts of the article, and approved the final draft.
- Ufuk Sönmez conceived and designed the experiments, performed the experiments, analyzed the data, prepared figures and/or tables, authored or reviewed drafts of the article, and approved the final draft.

### Human Ethics

The following information was supplied relating to ethical approvals (*i.e.*, approving body and any reference numbers):

The study received written authorization to adapt the PozQoL into Turkish through email communication with project team at La Trobe University where developed PozQoL. The authors have permission to use this instrument from the copyright holders (*Brown et al., 2018*). Additionally, ethical clearance to conduct the research was obtained from the ethics committee of Tepecik Training Hospital (approval date/number: 15.09.2022/2022/09-31) and the administration of the hospital where the participants were recruited. Informed consent from participants was taken.

### Data Availability

The raw data is available in the Supplemental Files.

### Clinical Trial Registration

The following information was supplied regarding Clinical Trial registration:

None.

### Supplemental Information

Supplemental information for this article can be found online at http://dx.doi.org/10.7717/peerj.17873#supplemental-information.

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
