# Peer review of "Validation of PozQoL scale in Turkish population living with HIV: a cross-cultural adaptation study"

_PeerJ, doi:10.7717/peerj.17873_

## Round 0.1 · original submission · Minor Revisions

The reviewers have requested some minor revisions to your manuscript. These revisions appear to be important but simple to do nonetheless. The reviewers have praised your manuscript for its overall ease to understand and its structure.

The reviewers have requested some additional clarity in the presentation of tables 1&3, as well as some clarification needed in the Methods section on the choice of statistical methods used.

Please read the reviewer's comments in full, this is not an exhaustive summary of their advice.

Reviewer 1 ·

Basic reporting

• Overall, the literature review seemed complete and well reasoned. It was easy to follow.
• English is good and clear.
• Abstract is clear and to the point.
• Table 1 and Table 3 were good, but they could be much better. Without knowing what the items of the measure were, just saying item 1, item 2, etc is not very meaningful. Putting the item stem would help a great deal in making interpretations. As a reader, this is not very meaningful to me unless I actually know what the item is.
• For table 1, I would still like to see what the factor loadings for the items where not significant. The authors could bold the ones that fit the factor and use standard lettering of the non-significant ones.
• Line 118 – Add “they” between “where developed”

Experimental design

• Study design is adequate and appropriate for this type of research. This is typical psychometric reserach.

Validity of the findings

• The conclusions fit the findings. Nothing more to add.

Additional comments

None.

Reviewer 2 ·

Basic reporting

1. The article uses clear and unambiguous, professional English.
2. The paper provides a comprehensive background on the topic, and cite relevant literature.
3. The article follows a professional structure, including sections for the introduction, methods, results, discussion, and conclusion. Tables and figures are shown.
4. The article is self-contained and the results are relevant and reasonable to the hypotheses.

Experimental design

1. The article presents original primary research within Aims and Scope of the journal.
2. The research question is clearly defined and relevant.
3. The investigation appears to be performed to a high technical and ethical standard. The use of translation-back-translation methodology and both exploratory and confirmatory factor analyses are appropriate and rigorous methods for validating the scale.
4. The methods section is generally detailed and provides sufficient information to allow replication of the study.
However, the study mentions recruiting 130 patients but does not detail how these participants were selected. Providing information on the sampling method (e.g., random sampling, convenience sampling) would minimize bias and improve the clarity and replicability of the study.
What's more, while the statistical methods used (EFA, CFA, Cronbach's alpha, ICC) are appropriate and well-explained. it's helpful to include the rationale for choosing specific cutoff values for fit indices (e.g., CFI > .95, RMSEA < .06) would add further clarity.

Validity of the findings

1. no comment.
2. The underlying data have been provided.
3. The conclusions are well stated and directly linked to the original research question, confirming the reliability and validity of the Turkish version of the PozQoL scale.

Additional comments

no comment

---

## Round 0.2 · accepted · Accept

Thank you for addressing the concerns and comments of the reviewers. Based on the advice of two reviewers, one who had not seen the previous versions of the manuscript and therefore was able to provide a fully independent review of the text, we will accept the manuscript for publication.

Reviewer 2 ·

Basic reporting

1. The manuscript is written in clear and professional English.
2. The article provides an adequate background and context.
3. The structure of the article follows standard scientific formats, including abstract, introduction, methods, results, discussion, and references.
4. Figures and tables are well-presented.
5. The manuscript is self-contained, presenting all necessary results to address the hypotheses.

Experimental design

1. The study offers original primary research through the cross-cultural adaptation and validation of the PozQoL scale.
2. The research question is well defined, relevant, and meaningful.
3. The investigation is thorough and follows rigorous methodological standards.
4. The methods are described in detail, allowing for replication.

Validity of the findings

1. The study presents a rigorous validation process for the Turkish version of the PozQoL scale, adhering to high technical and ethical standards.
2. All underlying data have been provided. They are robust, statistically sound, & controlled.
3. Conclusions are well stated.

·

Basic reporting

no comment

Experimental design

no comment

Validity of the findings

no comment

Additional comments

no comment